

# Quantifying the influence of mining dust particle deposition on the melting rate of nearby glaciers in northwestern China

Zhiyi Zhang[1], Xinyi Xu[2,3,*], Hideki Shimada[4], Wenfeng Wang[1,2,3,*], Xiaoyong Tong[5], Yuan Gao[1], and Weiming Guan[1]

[1]School of Geology and Mining Engineering, Xinjiang University, Urumqi 830046, China
[2]School of Resources and Geosciences, China University of Mining Technology, Xuzhou 221116, China
[3]Department of earth resources engineering, Kyushu University, Kyushu 819-0395, Japan
[4]Carbon Neutrality Institute, China University of Mining Technology, Xuzhou 221008, China
[5]The First Hydrology Engineering Geological Brigade of Xinjiang Bureau of Geology and Mineral Resources Geological Brigade, Xinjiang Bureau of Geo-exploration & Mineral Development, Urumqi 830000, China

*Correspondence to*: X. Xu (xinyixu@cumt.edu.cn), W. Wang (wenfwang@163.com)

**Abstract.** In addition to causing severe damage to human health and mechanical equipment, mineral dust particles (MDPs) also affect the rate at which glaciers melt. Although the acceleration of glacier melting by MDPs has attracted attention, there is limited understanding of the main controlling variables affected by MDPs that change the melting rate, and the mathematical relationships between each variable and the rate of melting remain to be fully elucidated. To address this problem, we first reconstructed the ablation environment to simulate changes in the rate of glacier melting under the influence of MDPs. The environment was analyzed through both physical and numerical experiments, and the response of glacier melting to multiple particles and individual particles on both macroscopic and microscopic levels was examined. Subsequently, based on thermodynamic laws, we theoretically derived a formula to calculate the increase in the rate of glacier melting attributable to MDPs. Through mutual validation of experiments and theory, we found that MDP coverage on the glacier surface increases the energy absorbed by the glacier, thereby resulting in an increased rate of melting, with an uplift of 10 %–40 %. The increase in the rate of melting is controlled primarily by four variables: particle number, particle diameter, irradiance, and particle surface albedo. Particle number, irradiance, and particle surface albedo each exhibit a linear relationship with the rate of increase in meltwater production, whereas particle diameter shows an exponential (quadratic) relationship. Our findings elucidate the mathematical relationship between MDPs and the rate of glacier melting, thereby providing scientific reference for glacier protection and accurate prediction of glacier melting rate.

## 1 Introduction

As an important freshwater resource, glaciers are known as "solid reservoirs" (Nie et al., 2021; Wagner et al., 2021; Zhang et al., 2022b). Changes in the volume of meltwater resulting from variation in the mass balance (accumulation and ablation) of glaciers have substantial impact on the ecological environment and human survival in downstream areas. For example, meltwater from glaciers in the Himalayas (Mir, 2021; Wood et al., 2020), Alps (Bearzot et al., 2023), and Andes (McCarthy





et al., 2022) is used for irrigating farmland, providing drinking water, and generating hydropower in South Asia, southern-central Europe, and some parts of South America. According to the World Glacier Monitoring Service, most monitored glaciers globally show a trend of shrinkage that indicates that glaciers are melting faster than before.

The phase transformation from ice to water might appear to be a straightforward process. However, the melting of glaciers is the outcome of a combination of factors. Generally, temperature plays the dominant role in glacier melting (Rounce et al., 2023), as evidenced by the shrinking of most glaciers worldwide following the global increase in surface air temperature. However, the rate of glacier melting is actually determined by the mass balance of the glacier, which depends on the physical state of the glacier and the environmental conditions, which include the surrounding temperature (Frans et al., 2018), glacier
morphology (area, slope) (Dobhal et al., 2021), glacier albedo (Dowson, Sirguey and Cullen, 2020; Zhang et al., 2021), solar radiation (Fyffe et al., 2021), precipitation (Zhang et al., 2022a), and surface wind velocity (Mandal et al., 2020). The dominance of temperature as a controlling factor can be diminished for some glaciers under the action of different conditions, or other influences could make greater contributions to glacier melting than they might have previously. For example, in alpine and polar regions, where the ambient temperature of glaciers is extremely low (usually below −10 °C), the contribution of
temperature to the overall rate of melting is weakened in comparison with that of other factors (Hock and Huss, 2021). In the Pamir region of the western Himalayas, where glaciers are strongly influenced by the monsoon, it has been reported that the contribution of the monsoon to glacier melting is currently greater than it has been in the past (Lone et al., 2022). Therefore, under specific conditions, there could be situations in which other factors make greater contributions to the rate of glacier melting. It is known that the rate of glacier melting is influenced primarily by the energy exchange state, which serves as the
fundamental controlling mechanism.

Solar radiation is one of the most important sources of energy in the process of glacier melting (Chen et al., 2021). It transfers energy directly to the glacier in the form of thermal radiation, and the amount of energy absorbed by the glacier is determined by the albedo (i.e., the ratio of incident to scattered light flux at the surface of an object) of its surface (Aubry-Wake, Bertoncini and Pomeroy, 2022; Singh et al., 2020). If the glacier is clean, its surface should have a high albedo. However, in its natural
state, a glacier is exposed to the open environment over a large area, making it very susceptible to pollution. Picard et al. (2020) found that the albedo of the surface of a clean glacier can be >0.9, whereas the albedo of a heavily polluted ice surface is only 0.1. Obviously, reduction in glacier surface albedo will increase the energy transferred to the glacier, which in turn will increase the rate of glacier melting.

Mineral dust particles (MDPs) have emerged as a major contributor to the accelerated rate of melting of glaciers. Large
quantities of mineral dust, comprising fine particles with high capacity for absorbing solar radiation, are typically produced during various open-pit mining operations that include blasting, excavation, crushing, loading, and transportation (Agboola et al., 2020; Tayebi-Khorami et al., 2019). Upon dispersal to the surface of glaciers by wind transport, MDPs substantial diminish the surface albedo, as observed on the Tianshan glaciers in Asia (Zhang et al., 2022b), Tibetan Plateau glaciers (Yan et al.,





2023), Andean glaciers (Cereceda-Balic et al., 2022), and Alpine glaciers (Oerlemans, Giesen and Van den Broeke, 2009),
thereby hastening glacier meltwater production. Thind et al. (2019) and Skiles et al. (2018) additionally noted the heightened
efficacy of MDPs in reducing the surface albedo of glaciers compared with that of other substances.

It is an unavoidable reality that the easily exploitable mineral resources are gradually depleting (Bardi, 2014; Diederen, 2009).
In the foreseeable future, resource extraction will expand to higher mountain ranges, intensifying the rate of glacier melting;
the Xinjiang region of China is already experiencing this situation (Fig. 1a). Despite notable advancements in current models
for predicting glacier melting, which incorporate factors such as glacier characteristics (Jennings and Hambrey, 2021),
topography (López-Moreno et al., 2020), climate (Liu et al., 2022), and hydrological features (Young et al., 2021), the impact
of MDPs might no longer be a random disturbing element in the glacier melting process but a primary controlling factor,
necessitating its inclusion as a crucial variable in glacier melting models. However, research on the energy transfer and flow
of MDPs during glacier melting is limited, and the direct mathematical relationship between MDPs and the rate of glacier
melting remains unclear. For example, Zhang et al. (2022b) explored the relationship and directional effects between MDPs
and melt rates based on in situ observations, which marked a significant advancement in research direction. Nevertheless, a
clear functional relationship was not established in their study. This ambiguity is likely to impact the accuracy of glacier
prediction models.

The primary objective of this study was to establish a mathematical relationship between MDPs and the rate of glacier melting.
To achieve this, we initially constructed a solar radiation simulation device and conducted physical simulation experiments to
observe the changes in meltwater when MDPs covered the ice surface. Subsequently, numerical simulation experiments were
conducted to monitor the energy transfer between MDPs and the glacier during the melting process by constructing a
thermodynamic field. These two experiments allowed comprehensive evaluation of the various factors related to MDPs during
glacier melting from both macroscale and microscale perspectives. Additionally, an energy equation for the migration of
mineral dust energy on the glacier surface was proposed. It is important to emphasize that our findings do not directly predict
the rate of glacier melting, but rather establish a mathematical relationship between MDPs and change in the rate of production
of glacier meltwater.

## 2 Materials and methods

### 2.1 Study area

The study area, located near the main ridge of the Tianshan Mountains (43°14'01.1440"N, 85°33'19.9535"E), is in the eastern
section of the Yilianhabir Ga Mountains in western Tianshan, Xinjiang Province, China (Fig. 1a). The area is characterized by
deep valleys and steep slopes with high-relief alpine topography. With elevations in the range 3160–4575 m, the area is
classified as a high-elevation region with perennial snow cover and a cold climate. An open-pit iron mine near a glacier is
located within this region (Fig. 1b). In the process of field investigation, we observed large numbers of foreign particles causing



serious pollution on the surface of some glaciers (Fig. 1c), and the level of pollution increased with proximity to the mining
      area. Comparison of World Imagery Wayback high-resolution satellite images revealed the ablation status of local glaciers
      over the past six years (Fig. 1d). It is evident that the glaciers within the study area experienced different degrees of retreat,
      with glaciers closer to the mining area retreating fastest. This suggests that the dark-colored MDPs on the surface of the glacier
      played a nonnegligible role in the rate of melting of the glaciers.

**Figure 1: (a)** Current distribution of glaciers and mineral deposits in the Xinjiang region of China (Zhang et al., 2023). **(b)** Aerial view of
the study area. **(c)** Glaciers and coverage of MDPs typical of the study area. **(d)** Retreat of Glacier #1 (near mining area) and Glacier #2 (far
from the mining area) during 2015–2021 (satellite images from World Imagery Wayback).

### 2.2 Samples and data

In October 2022, field operations undertaken in glacier #1 included in situ sampling of the glacier surface using a multipoint
      sampling technique (Fig. 1b). A geological hammer and clean shovel were used to obtain the required samples (each point
      sampling area of 50 × 50 cm), taking the surface glacier and its overlying snow layer. A total of 10 groups of samples were
      obtained, and the acquired samples were placed in the cryogenic sample box. After the samples were transported to the





laboratory, they were pretreated by means of standing and drying, after which they were sieved and graded using a standard
sieve, resulting in particles with diameter in the range 50–250 μm. In the laboratory, examination with a polarized light
microscope revealed that the MDPs had a predominant hemispherical morphological characteristic (Fig. 2a and Fig. 2b).
Further examination of the MDPs using X-ray diffraction revealed the composition of the particles and the proportion of each
constituent (Fig. 2c and Fig. 2d). Based on the X-ray diffraction results, it was determined that the particles on the surface of
the glacier comprised mainly fragments of skarn-type mineral rocks. This compositional similarity to the stripped rocks from
the adjacent mining area (Zhang et al., 2023; Zhang et al., 2022b) provides compelling evidence that the observed MDPs on
the glacier surface predominantly originate from mining-related activities. The density of such rocks is 2400–3500 kg m$^{-3}$, the
specific heat capacity is 750–1300 J kg$^{-1}$ K$^{-1}$, and the thermal conductivity is 1.5–7.0 W m$^{-1}$ K$^{-1}$ (Table A1).

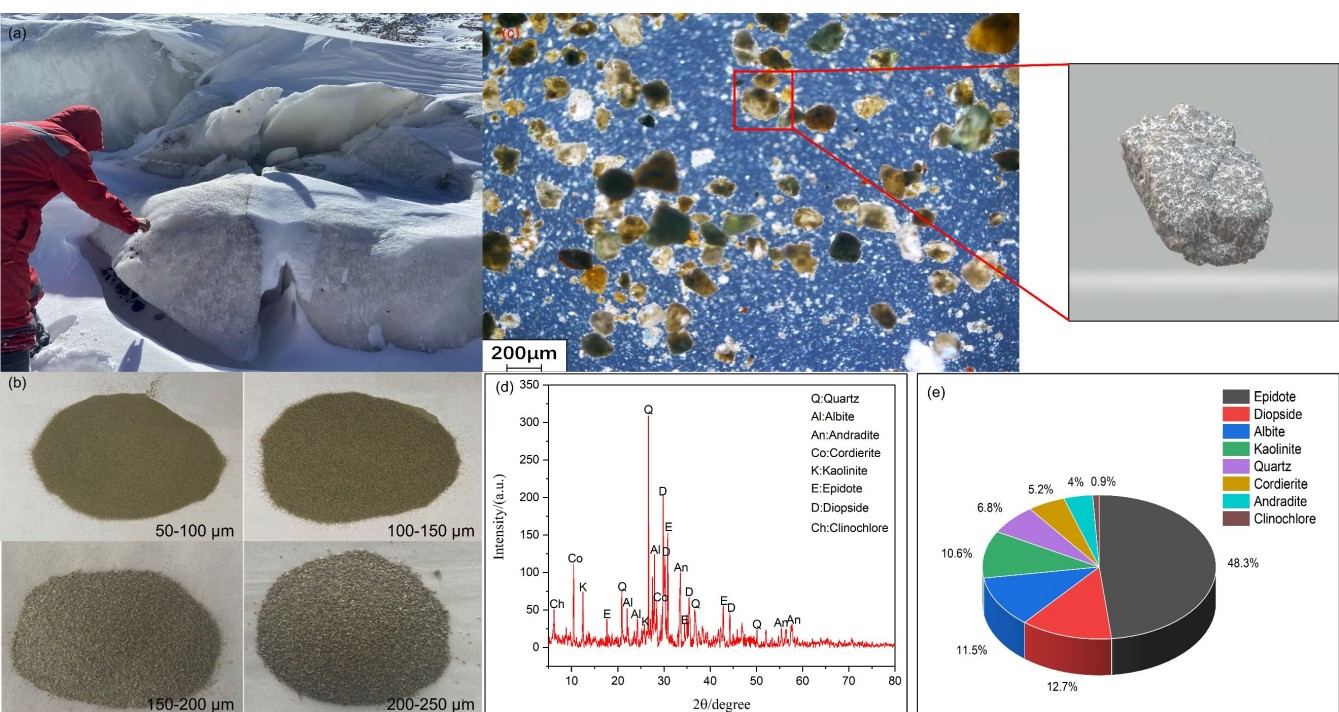

**Figure 2: (a)** Field collection of MDPs. **(b)** MDPs sorted into different diameter ranges after laboratory sieving. **(c)** Appearance of MDP
morphology after magnification under a polarized light microscope, 3D-model of an individual MDP. **(d)** Composition of substances
contained in the MDP samples. **(e)** Percentage of each substance in the MDP samples.

A fundamental fact is that earth's rotation causes day and night, while its tilted axis creates the Earth's season. Consequently,
the intensity of solar radiation on earth's surface changes constantly (Fig. 3a). For any location on earth's surface, the maximum
intensity of radiation received should be when the solar elevation angle is at its maximum (Fig. 3b). Taking the Northern
Hemisphere as an example, for the area north of the Tropic of Cancer (>23.26°N), the maximum solar elevation angle occurs
when the sun is directly above the Tropic of Cancer and the minimum solar elevation angle occurs when the sun is directly
above the Tropic of Capricorn. Based on the pattern of variation of solar irradiance at the surface of the earth, the range of
irradiance at a specific location can be determined. Using data from the National Aeronautics and Space Administration's



Clouds and the Earth's Radiant Energy System project, the hourly average solar irradiance throughout the day at these two

particular nodes in the study area was summarized for the period 2017–2021 (Table A2). It was found that the solar irradiance

during the same period in different years did not differ by more than 2 W m$^{-2}$. By averaging the observations for each period

during 2017–2021, it was concluded that the maximum radiation value in the study area occurred daily at 14:00–15:00 LT,

and that the maximum (minimum) value of annual solar irradiance was 1234 (547) W m$^{-2}$.



**Figure 3: (a)** Because of the great distance between the earth and the sun, we consider all rays of light projected onto the earth's surface as parallel light, with a difference in the angle of exposure to the sun at the same moment for different aspects of the earth. **(b)** If the energy radiated by the sun to the earth is quantized as an infinite number of beams of light, and assuming that each beam carries the same amount of energy, the density of energy received at the surface is different owing to differences in the angle of projection, i.e., $S_1 = S_2 > S_3$ (radiation intensity).

**2.3 Experiments**

Once MDPs cover the surface of a glacier, the original radiation absorption state of the glacier is altered, which leads to a

change in the rate of meltwater production. From the perspectives of thermodynamics and energy conservation, the reason





behind this macroscopic phenomenon should be that the original glacier energy balance has been altered. Therefore, we conducted two sets of experiments: the first set focused on assessing the effects of various parameters on the rate of production of glacier meltwater when the glacier surface was extensively covered with mineral dust, utilizing physical simulation experiments (Section 2.3.1); the second set of experiments examined the impact of various parameters of individual MDPs on energy absorption by the glacier during the melting process, utilizing numerical simulation experiments (Section 2.3.2). By adopting an experimental method that combined both macroscale and microscale perspectives, we could comprehensively study the impact of MDPs on glaciers.

### 2.3.1 Experiment #1: Glacier meltwater rate in physical simulations

To replicate the actual field conditions and analyze the effect of MDPs on the glacier meltwater rate, it was necessary to construct an experimental environment similar to the actual conditions observed in the field. Thus, the experimental environment involved construction of a solar radiation simulation device. This device used a xenon lamp as the radiation source, and the intensity and angle of irradiation could be adjusted arbitrarily (Fig. 4a and Fig. 4b). The xenon lamp was used as the radiation source to ensure that the source in the experiment had the same radiation effect as that of the sun. Among various artificial light sources, it has been shown that the spectral characteristics of a xenon lamp are closest to those of the sun (Fig. 4c), i.e., the spectrum of a xenon lamp is similar to that of the sun in the ultraviolet, visible, and near-infrared wavelength bands (Pottas et al., 2022).

Ice blocks of uniform size were obtained by injecting a specific amount of distilled water into molds (width × length × depth: 10 × 10 × 8 cm), which were then left to freeze. To ensure timely removal of meltwater, a drainage hole was created in the center of each ice block (Fig. 5). The square-shaped ice blocks offered a practical advantage in terms of obtaining a uniform distribution of MDPs and ensured ease of operation. Additionally, it was possible to ensure that the ice blocks received an equal level of surface irradiance, thereby promoting effective control of the variables in the study.

Because the angle of incidence of radiation received from the sun changes constantly throughout the day, theoretically, a physical simulation experiment should also adjust the irradiation angle with time. However, it was considered that short interruptions to the experiment to adjust the angle of irradiation would affect the results; therefore, three irradiation angles (i.e., 45°, 90°, and 135°) were selected for the physical simulation experiment. Given the effect of air temperature, it was considered that the duration of the experiment should be as short as possible, but with recognition that if the duration was too short or if the irradiance was too high, it would be impossible to accurately measure the subtle effects of the variables on the experimental results. After repeated pre-experiment tests, we opted for a duration of 20 min, with the ratio of time allocated to the three irradiance angles as 1:2:1.





**Figure 4: (a)** Design schematic of the solar radiation simulation device. By adjusting the magnitude of the radiation flux, the variation of irradiation in different seasons can be realized, and by adjusting the crawling speed controller, the angle of solar radiation at different times of the day can be realized. **(b)** The customized solar radiation simulation device used in the laboratory. **(c)** Comparison of the spectral characteristics of a xenon lamp (with a filter) and the sun, showing that the spectra are similar in all wavelength bands (Reber et al., 2005).



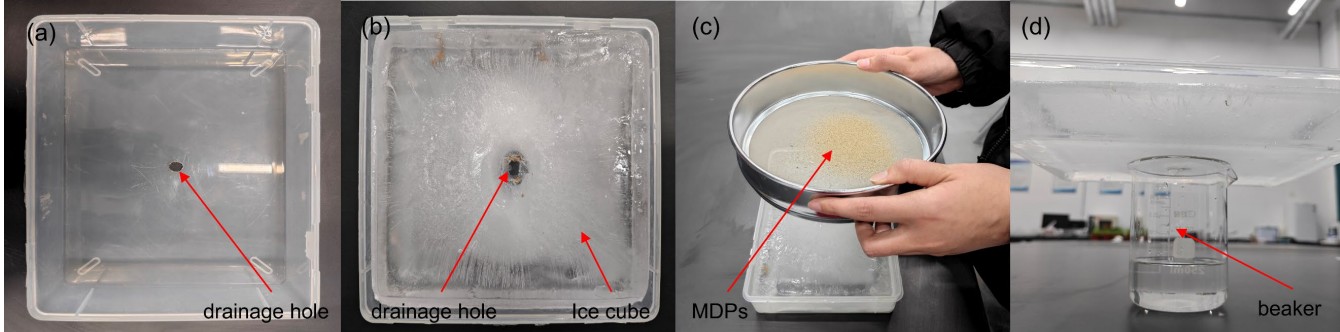

**Figure 5: (a)** The ice block mold (width × length × depth: 10 cm × 10 cm × 8 cm) with a drainage hole at the center. The drainage hole facilitates timely discharge of the meltwater. **(b)** An iron bar is placed at the center of the mold (for the drainage hole). A watertight seal between the iron bar and the mold is ensured, and then 500 mL of distilled water is injected into the mold. The mold is then placed in a freezer to create the ice block. **(c)** The MDPs are distributed evenly over the surface of the prepared ice blocks. **(d)** A beaker is placed centrally under the mold to collect meltwater discharged through the drainage hole.

Given the impact of the ambient temperature, which might lead to additional meltwater production on the lateral and bottom surfaces of the ice blocks, a control group was established. In the control group, MDPs were not applied to the surface of the ice blocks. Using Eq. (1), the meltwater volume exclusively attributed to MDPs can be determined:

$$V_{in} = V_{MDPs} - V_0 , \qquad\qquad (1)$$

where $V_{in}$ represents the meltwater volume solely associated with MDPs, $V_{MD}$ represents the amount of meltwater (mL) when the ice surface is covered with MDPs, and $V_0$ represents the meltwater amount (mL) in the absence of MDPs. The configurations of the various variables adopted for this experiment are presented in Table 1.

**Table 1.** Physics simulation experimental protocols

| Group number | 1 | 2 | 3 | 4 | 5 | 6 | 7 | 8 | 9 | 10 |
|---|---|---|---|---|---|---|---|---|---|---|
| Total mass of particles (g) | 0 | 0.5/1 | 0.5/1 | 0.5/1 | 0.5/1 | 0 | 0.5/1 | 0.5/1 | 0.5/1 | 0.5/1 |
| Irradiance (W/m²) | 400 | 400 | 400 | 400 | 400 | 600 | 600 | 600 | 600 | 600 |
| Particle diameter range (μm) | - | 50-100 | 100-150 | 150-200 | 200-250 | - | 50-100 | 100-150 | 150-200 | 200-250 |
| Group number | 11 | 12 | 13 | 14 | 15 | 16 | 17 | 18 | 19 | 20 |
| Total mass of particles (g) | 0 | 0.5/1 | 0.5/1 | 0.5/1 | 0.5/1 | 0 | 0.5/1 | 0.5/1 | 0.5/1 | 0.5/1 |
| Irradiance (W/m²) | 800 | 800 | 800 | 800 | 800 | 1000 | 1000 | 1000 | 1000 | 1000 |
| Particle diameter range (μm) | - | 50-100 | 100-150 | 150-200 | 200-250 | - | 50-100 | 100-150 | 150-200 | 200-250 |

*Note:* The room temperature at the time of the experiment was 15°C. In the control group, no MDPs were spread on the surface of the ice, so the total mass of the particles was "0" and the particle diameter was "-". " 0.5/1" then indicates that the total mass of MDPs spread on the surface of ice is 0.5 g and 1 g, respectively.





### 2.3.2 Experiment #2: Glacier energy absorption in numerical simulations


Finite element numerical analysis software widely employed in scientific research and engineering computations across various disciplines (Wu et al., 2023). In the present experiment, when viewed externally, the glacier and MDPs are subjected to solar radiation; however, internally, there is also potential for energy transfer between these entities, representing a composite continuous energy field involving two modes of thermal energy transfer—radiative heat transfer and solid heat

conduction. The use of the numerical analysis software allows construction of expansive models capturing terrestrial solar radiation on a macroscopic scale, while concurrently embedding microscopic models to account for solid thermal conduction.

A model of an individual MDP and the underlying glacier it covered was created in COMSOL Multiphysics. Then, a glacier with the same diameter but without the MDP was constructed to serve as a control (Fig. 6). The radiation field was constructed using the software-embedded earth–sun radiation model, which makes it very easy to directly input irradiance and the latitude

and longitude of the irradiated area. The solid heat transfer module was used for the portion of the particle in contact with the underlying glacier. The duration of irradiation adopted for the numerical simulation was taken as the average duration of the day when the sun was directly above the Tropic of Cancer over the five studied years.

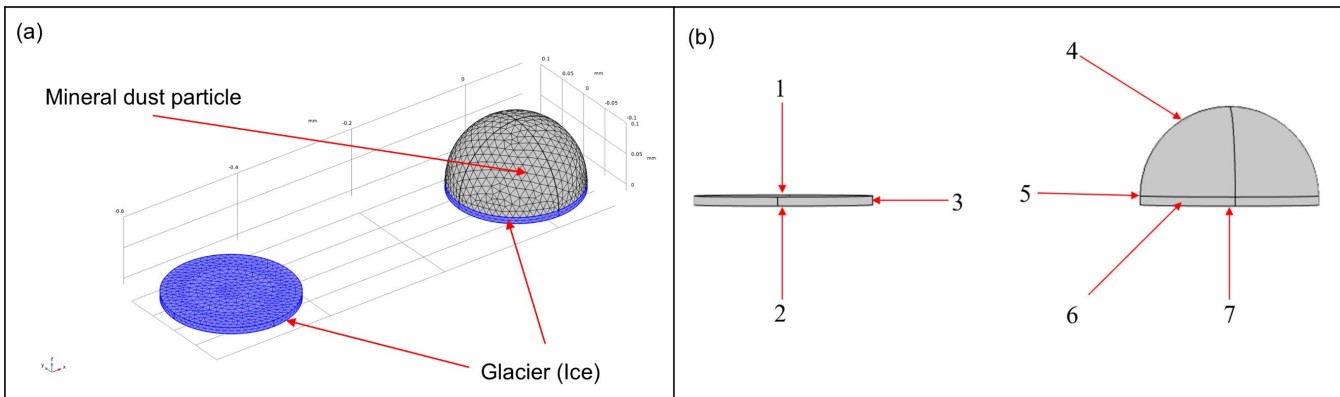

**Figure 6: (a)** The model created in COMSOL Multiphysics. **(b)** The boundary settings of the model: **(b-1)** represents the diffuse reflection
boundary of the glacier, serving as the boundary at which the glacier receives radiation; **(b-2)** represents the constant temperature boundary of the glacier, serving as the isothermal layer connected to the lower glacier; **(b-3)** represents the thermally insulated boundary of the glacier, serving as the continuous boundary laterally connected to the glacier; **(b-4)** represents the diffuse reflection boundary of the MDP, serving as the boundary at which the MDP receives radiation; **(b-5)** represents the transitional boundary, serving as the transitional boundary at which the MDP interfaces with the glacier; **(b-6)** serves the same function as (b-3); and **(b-7)** serves the same function as (b-2).

Surface emissivity is a crucial parameter in radiative heat transfer, serving as an important metric for measuring the ability of an object to absorb radiation (Howell et al., 2020). It is a dimensionless ratio with a numeric range of 0–1, where a value closer to 1 indicates closer resemblance to a blackbody (capable of fully absorbing radiant energy). A coincidental case in radiative heat transfer theory is that, if an object is a gray body with no transmissivity, the condition "1 − albedo = emissivity" applies (Bejan, 2016). It can be determined from Fig. 2a and Fig. 2b that MDPs are characterized broadly by dark colors and a relatively

rough surface. Surface emissivity is influenced directly by factors such as the color and material properties of the surface of the receiving object. According to relevant studies on rock minerals, the emissivity of most dark-colored rocks is generally >0.9





(Mineo and Pappalardo, 2021; Sass et al., 2023). Therefore, from the perspective of exploring the laws in this study, the surface emissivity of the MDPs can be set to any value above 0.9 in the model, and it is entirely possible to use albedo instead of emissivity. In the numerical simulation experiments, we set the surface emissivity to 0.92, 0.94, 0.96, and 0.98, corresponding

to albedo values of 0.08, 0.06, 0.04, and 0.02, respectively.

It is evident from the formula for the phase transition energy (Eq. (2)) governing the conversion of ice to water that the quantity of glacier meltwater is directly proportional to the energy absorbed. By monitoring the variations in the energy absorption by glaciers, changes in the rate of melting can be determined indirectly. In the experimental procedure, real-time changes in heat flux were obtained using probes embedded in the software. Simultaneously, the magnitude of the transferred energy was

calculated using Eq. (3). According to Eq. (2) and Eq. (4), it is evident that the increment in the rate of production of meltwater is determined by the injected energy. This is because the latent heat of fusion in the phase change from ice to water is a constant of physics, and water density is also a constant of physics. Consequently, a direct proportional relationship exists between the increment of the rate of meltwater production and the increment of injected energy. Thus, the correspondence between the increment in the meltwater rate and the variables aligns with the correspondence between the increment of injected energy and

the respective variables. The experiment conducted in this study involved the selection of three independent variables, i.e., particle diameter, albedo ("$1 - \text{emissivity}$"), and solar irradiance, each with four levels. A comprehensive set of 64 experiments was conducted (Table 2) to systematically analyze the influence of these three independent variables on the energy absorption by glaciers.

$$Q_w = m_G \cdot L_f \, , \tag{2}$$

$$Q_e = \int_0^t (q_1 - q_2) \cdot S \, dt \, , \tag{3}$$

$$V = \frac{m}{\rho} \, , \tag{4}$$

where $Q_w$ is the absorbed heat (J); $m_G$ is the mass of the substance (g); $L_f$ is the latent heat of fusion, which is the amount of

heat required to convert a unit mass of the substance from the solid phase to the liquid phase (for water, $L_f$ is approximately 334 J g$^{-1}$); $Q_e$ is the increase in energy value due to the presence of MDPs; $q_1$ represents the magnitude of the heat flux conducted by the particles to the lower boundary of the ice surface; $q_2$ denotes the magnitude of the radiant heat flux conducted downward from the ice surface when there is no particle coverage; and $S$ refers to the magnitude of the area of the ice surface covered by MDPs; $V$ represents the volume of meltwater (m$^3$), $m$ is the mass of the glacier or ice block (Kg), and $\rho$ denotes the

density of water (Kg m$^3$).



**Table 2.** Numerical simulation experimental protocols

| Group number | 1 | 2 | 3 | 4 | 5 | 6 | 7 | 8 | 9 | 10 | 11 | 12 | 13 | 14 | 15 | 16 |
|---|---|---|---|---|---|---|---|---|---|---|---|---|---|---|---|---|
| Solar irradiance (W/m$^2$) | 1300 | 1067 | 833 | 600 | 1300 | 1067 | 833 | 600 | 1300 | 1067 | 833 | 600 | 1300 | 1067 | 833 | 600 |
| Particle diameter (μm) | 200 | 200 | 200 | 200 | 200 | 200 | 200 | 200 | 200 | 200 | 200 | 200 | 200 | 200 | 200 | 200 |
| Albedo | 0.02 | 0.02 | 0.02 | 0.02 | 0.04 | 0.04 | 0.04 | 0.04 | 0.04 | 0.06 | 0.06 | 0.06 | 0.08 | 0.08 | 0.08 | 0.08 |
| Group number | 17 | 18 | 19 | 20 | 21 | 22 | 23 | 24 | 25 | 26 | 27 | 28 | 29 | 30 | 31 | 32 |
| Solar irradiance (W/m$^2$) | 1300 | 1067 | 833 | 600 | 1300 | 1067 | 833 | 600 | 1300 | 1067 | 833 | 600 | 1300 | 1067 | 833 | 600 |
| Particle diameter (μm) | 150 | 150 | 150 | 150 | 150 | 150 | 150 | 150 | 150 | 150 | 150 | 150 | 150 | 150 | 150 | 150 |
| Albedo | 0.02 | 0.02 | 0.02 | 0.02 | 0.04 | 0.04 | 0.04 | 0.04 | 0.04 | 0.06 | 0.06 | 0.06 | 0.08 | 0.08 | 0.08 | 0.08 |
| Group number | 33 | 34 | 35 | 36 | 37 | 38 | 39 | 40 | 41 | 42 | 43 | 44 | 45 | 46 | 47 | 48 |
| Solar irradiance (W/m$^2$) | 1300 | 1067 | 833 | 600 | 1300 | 1067 | 833 | 600 | 1300 | 1067 | 833 | 600 | 1300 | 1067 | 833 | 600 |
| Particle diameter (μm) | 100 | 100 | 100 | 100 | 100 | 100 | 100 | 100 | 100 | 100 | 100 | 100 | 100 | 100 | 100 | 100 |
| Albedo | 0.02 | 0.02 | 0.02 | 0.02 | 0.04 | 0.04 | 0.04 | 0.04 | 0.04 | 0.06 | 0.06 | 0.06 | 0.08 | 0.08 | 0.08 | 0.08 |
| Group number | 49 | 50 | 51 | 52 | 53 | 54 | 55 | 56 | 57 | 58 | 59 | 60 | 61 | 62 | 63 | 64 |
| Solar irradiance (W/m$^2$) | 1300 | 1067 | 833 | 600 | 1300 | 1067 | 833 | 600 | 1300 | 1067 | 833 | 600 | 1300 | 1067 | 833 | 600 |
| Particle diameter (μm) | 50 | 50 | 50 | 50 | 50 | 50 | 50 | 50 | 50 | 50 | 50 | 50 | 50 | 50 | 50 | 50 |
| Albedo | 0.02 | 0.02 | 0.02 | 0.02 | 0.04 | 0.04 | 0.04 | 0.04 | 0.04 | 0.06 | 0.06 | 0.06 | 0.08 | 0.08 | 0.08 | 0.08 |

## 3 Results

### 3.1 Relationship between glacier meltwater and MDPs in the physical simulations

Following the design of our physical simulation experiment, we first recorded baseline meltwater values (free MDPs) for the

control group at different irradiance levels (Fig. 7a). In ascending order of irradiance, the corresponding meltwater volumes were 72 mL, 89 mL, 109 mL and 130 mL. Subsequently, we used Eq. (2) to process the data from the experimental and control groups, obtaining the variations in meltwater increase under the influence of MDPs (Fig. 7b). When MDPs participate in the glacier melting process, the meltwater rate increases, and the extent of this enhancement is directly proportional to the mass of the MDPs. The values of meltwater increase shown in the figure are all positive, indicating that regardless of changes in

irradiance and MDP size, the presence of MDPs enhances the meltwater rate (with meltwater volume serving as a proxy for the meltwater rate given equal experimental durations). However, the extent of this enhancement varied under different conditions, with the increase in meltwater volume in the range 6–50 mL, i.e., the meltwater rate increased by approximately 10–40 %. Comparison of the meltwater volume corresponding to coverage by 0.5- and 1-g masses of MDPs revealed that under identical conditions of particle size and irradiance on the surface of the ice blocks, the meltwater volume associated with

1-g MDP coverage was approximately double that associated with 0.5-g MDP coverage. Furthermore, the error between the doubled meltwater volume for 0.5-g MDP coverage and 1-g MDP coverage was typically in the range 1–2 mL.




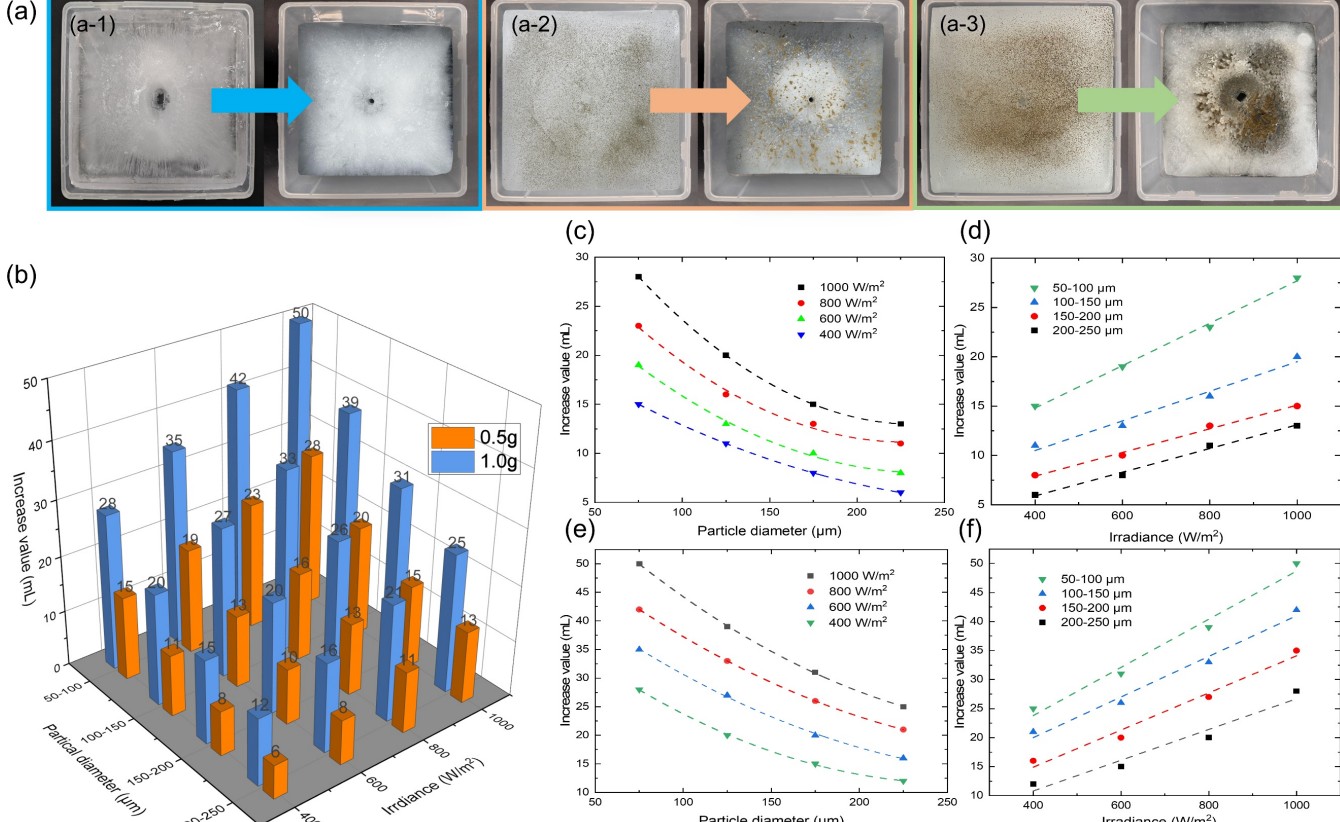

**Figure 7: (a)** Photographs of the initial and final states of some ice blocks taken during the experiment (a-1: free-MDPs; a-2 and a-3: 0.5- and 1-g mass of MDPs). **(b)** Total increase in meltwater volume associated with coverage by 0.5- and 1.0-g masses of MDPs on the surface of the ice block. **(c)** and **(e)** Change in meltwater volume associated with particle diameter for coverage by 0.5- and 1-g masses of MDPs, respectively, at different irradiances; **(d)** and **(f)** change in meltwater volume associated with irradiance for coverage by 0.5- and 1-g masses of MDPs, respectively, at different particle diameters. Dashed lines in **(c–f)** represent the fitted curves.

At constant irradiance and mass, the larger the particle diameter and the smaller the meltwater rate, and the two show a clear square relationship (Fig. 7c and Fig. 7e). By fitting the meltwater volumes corresponding to different particle diameters at the same irradiance, it can be seen intuitively that the meltwater volume presents a broadly squared decreasing relationship with particle diameter. Longitudinal comparisons (different irradiance levels) further show that the meltwater rate corresponding to particles with higher irradiance have larger base values.

Under the condition of constant particle diameter and mass, the higher the irradiance and the higher the meltwater rate, and the two show an obvious positive proportional relationship (Fig. 7d and Fig. 7f). By fitting the meltwater volume corresponding to different irradiance at the same particle size, it can be seen intuitively that the meltwater volume presents a broadly linear trend of increase with irradiance diameter. Longitudinal comparisons (different particle diameter sizes) further show that the meltwater rate corresponding to smaller particle sizes have larger base values.

 

## 3.2 Relationship between glacier energy intake and MDPs in the numerical simulations

Following the design of our numerical simulation experiments, the value of the increase in glacier energy intake associated
with the presence of MDPs can be calculated directly using Eq. (3). In terms of the value of the increase in energy intake versus
the change in irradiance, for the same particle diameter and albedo, the increase in the value of glacier energy intake increases
with irradiance (Fig. 8a). From the fitted curves, it is evident that the increase in irradiance exhibited a linear relationship with
the increase in energy absorbed by the glacier. Specifically, the relationship between irradiance (designated as w) and the
increase in energy absorbed by the glacier (designated as y) can be expressed as $y = \beta + \gamma \cdot w$. This functional relationship was
also observed in other experimental groups (Fig. 8b–d), indicating a consistent pattern. Moreover, from Fig. 8a–d, it is evident
that when the particle diameter is the same, the increased energy absorbed is determined by the particle surface albedo (within
the same group), i.e., the smaller the particle surface albedo, the larger the base coefficient (β) and (γ). When the particle
surface albedo is the same, the increased energy absorbed is determined by the particle diameter (within different groups), i.e.,
the larger the particle diameter, the larger the base coefficient (β) and (γ).

In terms of the value of the increase in energy intake versus the change in particle diameter, for the same irradiance and particle
surface albedo, the increase in the value of the glacier energy intake increases with particle diameter (Fig. 8e). From the fitted
curves, it is evident that the increase in particle diameter is exponentially related to the increase in glacier energy intake, with
an exponent value of approximately 2. This means than the functional relationship between particle diameter (this independent
variable is set as x) and the increase in glacier energy intake (this dependent variable is set as y) can be expressed as $y = \alpha \cdot x^2$,
which was also found in the other experimental groups (Fig. 8f–8h). Furthermore, from Fig. 8e–h, it is evident that for the
same irradiance, the increased energy absorbed is determined by the particle surface albedo (within the same group), i.e., the
larger the particle surface albedo, the larger the base coefficient (α). Similarly, for the same particle surface albedo (within
different groups), the increased energy absorbed is determined by the irradiance, i.e., the greater the irradiance, the greater the
base coefficient (α).

In terms of the value of the increase in energy intake versus the change in particle surface albedo, for the same particle diameter
and irradiance, the increase in the value of glacier energy intake decreases with the increase in particle surface albedo (Fig. 8i).
From the fitted curves, it is evident that the increase in particle surface albedo is linearly related to the reduction in energy
absorbed by the glacier, meaning that the particle surface albedo (the independent variable denoted as v) and the reduction in
energy absorbed by the glacier (denoted as y) can be expressed as $y = \delta + \eta \cdot v$. This functional relationship was also observed
in the other experimental groups (Fig. 8j–l). Additionally, from Fig. 8i–l, it is evident that for the same particle diameter, the
increase in energy is determined by the irradiance (within the same groups), with higher solar irradiance resulting in larger
base coefficients (δ) and (η). For the same irradiance, the increase in energy is determined by the particle diameter (within
different groups), with larger particle diameters resulting in larger base coefficients (δ) and (η).





**Figure 8: (a–d)** Changes between particle diameter and the value of increase in energy intake by glaciers. **(e–h)** Changes between irradiance and the value of increased energy intake by glaciers. **(i–l)** Changes between surface albedo and the increased value of glacier energy intake.

## 4 Discussion

### 4.1 Roles of variables in enhancing the meltwater rate

The set of physical experiments and the set of simulation experiments were both designed to examine the effects of particle diameter and irradiance. The difference is that the physical experiments included particle mass as an additional variable, while the simulations introduced particle surface albedo as a variable. Following the order of the variables mentioned above, we discuss the role of each variable in increasing the rate of meltwater production.

First, the increase in meltwater rate exhibits a quadratic relationship with particle diameter, satisfying the functional relationship: $y = ax^2 + bx + c$ (where $y$ is the increase in meltwater rate and $x$ represents the particle diameter). For the results



of both the physical experiments (Fig. 7c and Fig. 7e) and the numerical experiments (Fig. 8e–h), standard quadratic curves were obtained through fitting, yet with opposite trends. While these results might appear contradictory, they actually represent two different expressions of the same objective reality. In the physical experiments, the total mass of MDPs was controlled. With the total mass unchanged, smaller particle diameters result in a greater number of particles, thereby increasing the actual surface area receiving radiation. In the numerical experiments, the focus was on individual MDPs, and a particle with a larger

diameter naturally results in a larger surface area for receiving radiation. Therefore, the increase in the melting rate was determined by the size of the radiation-receiving surface area of the MDPs, which depends on both the diameter and the quantity of the MDPs.

Second, the increase in the meltwater rate exhibits a linear relationship with irradiance, following the functional relationship $y = ax + b$ (where $y$ is the increase in meltwater rate and $x$ is solar irradiance). This relationship was confirmed in both the

physical experiments (Fig. 7d and Fig. 7f) and the numerical experiments (Fig. 8a–d). This is primarily because, under equivalent conditions, the level of irradiance determines the total radiative energy received by the glacier (Qi et al., 2022; Singh et al., 2020). An increase in irradiance effectively intensifies the overall glacier energy input, with the MDPs acting as a "channel" or "transmission medium" in this process.

Third, the increase in the meltwater rate is linearly related to mass, satisfying the functional relationship $y = ax + b$ (where $y$

is the increase in meltwater rate and $x$ is the mass of the MDPs). In the physical experiments (Fig. 7b), the meltwater rate associated with 1-g MDP coverage was double that associated with 0.5-g MDP coverage mass. This is because controlling the mass of the MDPs essentially controls the particle numbers or surface area for receiving radiation. When the mass doubles, the corresponding number of particles or radiation-receiving area also doubles. As a medium for energy transfer, the number of MDPs has substantial impact on the energy absorption of the entire glacier.

Finally, the increase in the meltwater rate is linearly related to particle surface albedo, following the functional relationship $y = ax + b$ (where $y$ is the increase in meltwater rate and $x$ is the particle surface albedo). It is worth noting that the albedo of the particle surface has an effect on the rate of increase in glacier melting that is the opposite of the effect of the three variables discussed above. For incoming radiation reaching the particle surface, a higher particle surface albedo means less energy is absorbed because albedo directly determines the intensity of radiation absorption (Howell et al., 2020). In the simulation

experiments (Fig. 8i–l), the relationship between surface albedo and glacier energy absorption is directly reflected. During radiation, surface albedo acts as a "valve" controlling the flow rate, thereby determining the efficiency of absorption of solar radiation by the glacier.

## 4.2 Thermodynamic basis of MDP-accelerated melting

When the original ice and snow surface of a glacier is replaced by MDPs, the particles act as an opaque medium, impeding

direct absorption of solar radiation by the ice and snow of the glacier. Instead, the particles themselves absorb the incoming





solar radiation, thereby becoming the primary absorbers (Fig. 9). Under constant external irradiance conditions, the absorption of energy depends primarily on the surface area and albedo of the MDPs. Most MDPs present on the glacier surface have low albedo (Li et al., 2021b; Tuzet et al., 2020), and regardless of their surface morphology, they exhibit complex three-dimensional structures that substantially increase the radiative surface area. Because MDPs are mostly transported to the glacier surface by

wind, those particles reaching the glacier surface typically have small mass and volume (Chouaib and Caissie, 2021; Xiong et al., 2022). The inherent properties of the MDPs, including lower specific heat capacity and higher thermal conductivity (Table A1), enable them to rapidly transfer absorbed energy to the contacting glacier. The combined influence of these three factors provides the thermodynamic basis for MDPs to expedite increased glacier melting.

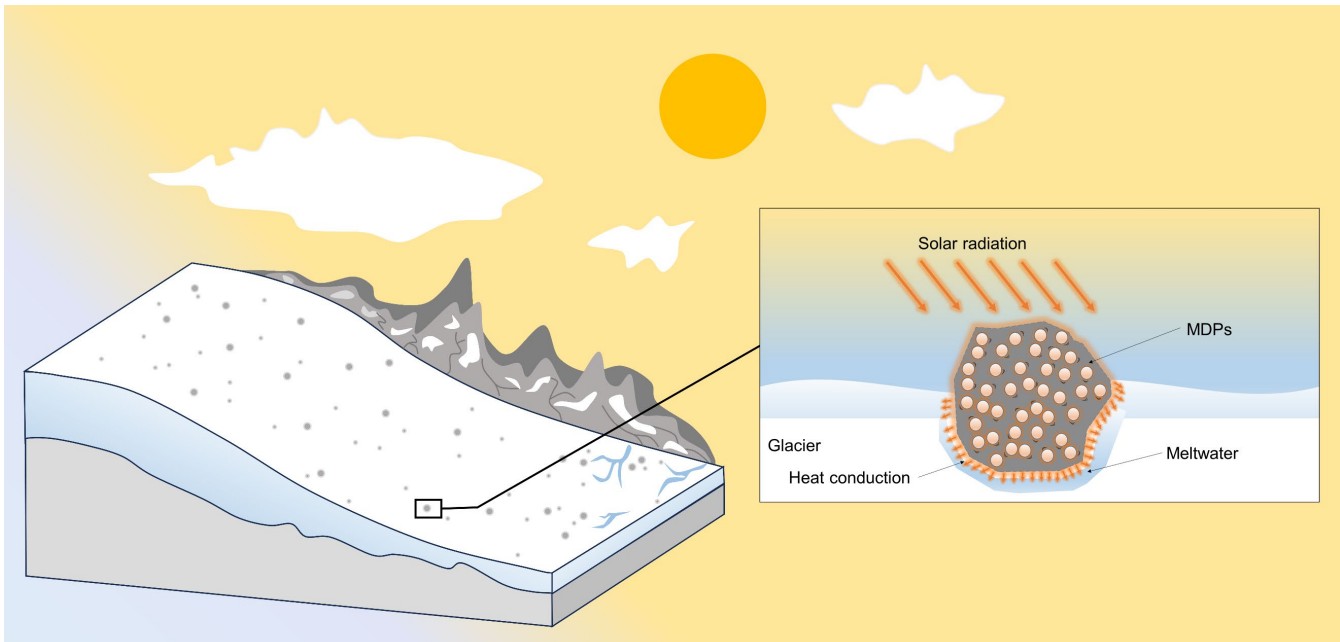

**Figure 9:** Under the influence of solar radiation, MDPs continuously absorb radiant heat energy from the surface layer, accelerating their own molecular motion (generating thermal energy). Eventually, this thermal energy is transferred to the underlying contacting glacier, leading to the generation of meltwater.

In an ideal state, it can be assumed that the average irradiance on the surface of a glacier in a certain location during a certain period is ($q$), that a particle of mineral dust with diameter $d_{MDP}$ and particle surface albedo of $\alpha_{MDP}$ is present, while disregarding

uncontrollable factors such as cloud cover, the albedo of the glacier surface is $\alpha_G$, and the duration of radiation exposure is ($t$). Under these conditions, the radiation received by the particle can be calculated as follows:

$$S_1(d) = 4\pi R^2 \cdot \frac{1}{2} = \frac{\pi}{2} \cdot d_{MDP}^2 \ . \tag{5}$$

The heat radiation absorbed by the particle can be represented by the following:





$$Q_1 = q \cdot S \cdot t = (1 - \alpha_{MDP}) \cdot q \cdot \frac{\pi}{2} \cdot d_{MDP}^2 \cdot t \; . \tag{6}$$

After receiving solar radiation, the particles will gradually transfer energy downward until the energy is eventually transferred to the underlying ice surface. Owing to the short duration of energy transfer through the particles (because they are small), the

energy dissipated within them can be neglected in the thermal conduction process. Thus, the energy $Q_2$ transferred through particle thermal conduction is equivalent to $Q_1$, which can be represented by the following:

$$Q_2 = Q_1 \; . \tag{7}$$

When covered by particles, the glacier has greater energy intake. However, the amount of direct radiation energy is also reduced, which must be subtracted for accurate calculation. The radiation energy received over an area of the glacier surface with diameter equal to that of the particles is as follows:

$$S_0(d) = \pi R^2 = \frac{\pi}{4} d_{MDP}^2 \; , \tag{8}$$

The amount of heat radiation that would have been absorbed can be derived as follows:

$$Q_0 = q \cdot s \cdot t = (1 - \alpha_G) \cdot q \cdot \frac{\pi}{4} \cdot d_{MDP}^2 \cdot t \; . \tag{9}$$

The increased energy can be expressed as follows:

$$\Delta Q = Q_2 - Q_0 = \frac{\pi}{4} \cdot d_{MDP}^2 \cdot q \cdot t \cdot \left( 2(1 - \alpha_{MDP}) - (1 - \alpha_G) \right) . \tag{10}$$

If there are (n) particles with equal diameter covering the glacier surface, the additional energy added attributable to the covering of particles can be represented by the following:

$$\Delta Q = \sum_1^n \frac{\pi}{4} \cdot d_{MDP}^2 \cdot q \cdot t \cdot \left( 2(1 - \alpha_{MDP}) - (1 - \alpha_G) \right) . \tag{11}$$

This relationship can be further explored to examine the relationship between energy and meltwater. By combining Eq. (2),

Eq. (4), and Eq. (11), the increase in the volume of meltwater generated because of the MDPs can be expressed as follows:



$$V = \frac{m}{\rho} = \frac{1}{\rho} \cdot \frac{Q_w}{L_f} = \frac{1}{\rho} \cdot \frac{\Delta Q}{L_f} = \sum_1^n \frac{d_{MDP}^2 \cdot q \cdot t \cdot (2(1-\alpha_{MDP})-(1-\alpha_G))}{4L_f \cdot \rho} \cdot \quad (12)$$

The increase in the meltwater rate can then be represented by the following:

$$v = \frac{V}{t} = \sum_1^n \frac{d_{MDP}^2 \cdot q \cdot \left(2(1-\alpha_{MDP})-(1-\alpha_G)\right)}{4L_f \cdot \rho}, \quad (13)$$

where $v$ is the increase of the rate of meltwater production (m$^3$ s$^{-1}$), $t$ is the duration of irradiation (s), $d_{MDP}$ is the radius of the MDPs (m), $q$ is the magnitude of irradiance (W m$^{-2}$), and $\rho$ is the density of water (kg m$^{-3}$).

If the glacier surface contains other MDPs with different properties (e.g., particle radius and surface albedo), Eq. (13) can be extended as follows:

$$v = \sum_1^n \frac{d_{MDP1}^2 \cdot q \cdot (2(1-\alpha_{MDP1})-(1-\alpha_G))}{4L_f \cdot \rho} + \sum_1^n \frac{d_{MDP1}^2 \cdot q \cdot (2(1-\alpha_{MDP2})-(1-\alpha_G))}{4L_f \cdot \rho} + \sum_1^n \frac{d_{MDP1}^2 \cdot q \cdot (2(1-\alpha_{MDP3})-(1-\alpha_G))}{4L_f \cdot \rho} + \quad (14)$$
$$\cdots \sum_1^n \frac{d_{MDP2}^2 \cdot q \cdot (2(1-\alpha_{MDP1})-(1-\alpha_G))}{4L_f \cdot \rho} + \sum_1^n \frac{d_{MDP2}^2 \cdot q \cdot (2(1-\alpha_{MDP2})-(1-\alpha_G))}{4L_f \cdot \rho} + \sum_1^n \frac{d_{MDP2}^2 \cdot q \cdot (2(1-\alpha_{MDP3})-(1-\alpha_G))}{4L_f \cdot \rho} +$$
$$\cdots \sum_1^n \frac{d_{MDP3}^2 \cdot q \cdot (2(1-\alpha_{MDP1})-(1-\alpha_G))}{4L_f \cdot \rho} + \sum_1^n \frac{d_{MDP3}^2 \cdot q \cdot (2(1-\alpha_{MDP2})-(1-\alpha_G))}{4L_f \cdot \rho} + \sum_1^n \frac{d_{MDP3}^2 \cdot q \cdot (2(1-\alpha_{MDP3})-(1-\alpha_G))}{4L_f \cdot \rho} + \cdots \cdot$$

Rearrangement of Eq. 14 yields the following:

$$v = \sum_{1,1,1}^{n,i,j} \frac{\pi \cdot d_{MDPi}^2 \cdot q \cdot (2(1-\alpha_{MDPi})-(1-\alpha_G))}{4L_f \cdot \rho} \cdot \quad (15)$$

Comparison of the results of the experiments with theoretical derivations revealed that the relationship between the increase in the rate of meltwater production and the various variables can be mutually validated, showing reasonable consistency. Based on calculation of the increase in meltwater rate using Eq. (15), it is observed that the increase is linearly proportional to particle mass, irradiance, and particle surface albedo, while being proportionally related to the square of the particle diameter. This aligns perfectly with the results of our experiments.

## 4.3 Applicability of the formula and other possible variables

While two distinct experimental approaches were employed in this study to integrate the macroscale and microscale dimensions and unify them with theoretical derivations, there are inherent limitations in the energy equation derived for the increase in the rate of meltwater production. This equation is applicable when the MDPs on the glacier surface are not stacked. However, in cases where MDPs are stacked or have multiple layers, voids will inevitably be present between particles. The



poor thermal conductivity of air (Najafi-Silab et al., 2023) within these voids could substantially affect heat transfer. Additionally, the size of the contact surface between particles cannot be established accurately, thereby affecting the overall determination of the heat transfer efficiency. Therefore, if the MDPs on the glacier surface were stacked, the theoretical increase in the rate of meltwater production might exceed the actual increase.

Wind speed might also influence the transfer of energy in relation to MDPs on the glacier surface. According to the general principles of thermodynamics, MDPs, after absorbing solar radiation, initially elevate their own thermal energy. When wind flows over the particle surface, the convective heat transfer of air increases the dissipation of energy absorbed by the particles, inevitably reducing the effective transfer of particle thermal energy to the glacier. However, previous studies indicated that a higher glacier surface flow rate corresponds to a faster rate of glacier melting (Bhushan et al., 2018; Laffin et al., 2023; Wu et al., 2020; Zhou, Chen and Cheng, 2021). This discrepancy might introduce bias in assessment of the impact of MDPs on the rate of glacier meltwater production. Consequently, in our experimental process, we excluded the factor of wind speed. Nevertheless, wind speed is likely to be a crucial factor that merits further exploration.

### 4.4 Representation of the research and new focus on glacier conservation

The glacial environment within our study area exhibits strong representativeness. First, on the global scale, almost all glaciers have a surface contaminated by MDPs, including the glaciers in the Third Pole region (Li et al., 2022; Li et al., 2021b), Greenland (Bohn et al., 2022; Cintron-Rodriguez et al., 2022), and the Arctic (Vérin et al., 2022). This indicates that the glacial ablation environment within our study area is not an exception but rather an example of a widespread phenomenon. Second, under the influence of wind, some MDPs can disperse over thousands of miles before settling on the surface of a glacier (Gelman Constantin et al., 2020; Li et al., 2021a; Zhang et al., 2023), suggesting that even glaciers distant from sources of MDP pollution might be affected. Furthermore, the MDPs used in our experiments are not examples of particles with the highest solar radiation absorption capability; materials such as black carbon demonstrate much stronger radiation absorption ability (Gul et al., 2021; Kang et al., 2020), thereby leading to more pronounced increase in the rate of glacier meltwater production. Therefore, the external substances selected for intervention on glacial surfaces in this study are highly typical. Clarifying their relationship with the rate of glacier meltwater production is highly meaningful and can provide insights regarding other interfering substances with different physicochemical properties.

Whether considering the current global trends in glacial retreat or evaluating the stimulating effect of MDPs on the rate of meltwater production in our study, the impact of MDPs on glacier melting demands attention. The findings of our quantitative study of MDPs could help further refine predictions of the rate of glacier melting, thereby facilitating enhanced glacier protection. Simultaneously, we aspire to contribute theoretical insights to support the development of emission control standards for MDPs.



## 5 Conclusions

This study employed a combination of physical and numerical experiments to quantitatively investigate the promoting effect of MDPs on the rate of glacier melting. By measuring the volume of meltwater in a beaker, the study macroscopically analyzed

the impact of particle number, irradiance, and particle diameter on the rate of increase in meltwater production. Through monitoring the energy transfer at the particle–glacier interface, the study elucidated, at the microscopic scale, the influences of irradiance, particle diameter, and surface albedo on the rate of increase in meltwater production. Based on macroscopic and microscopic studies, the research was extended to general scenarios through thermodynamic theoretical derivations, and a universal energy formula was derived to quantify the accelerated rate of glacier melting caused by MDPs. The main findings

of the study are as follows.

(1) The presence of MDPs on the glacier surface accelerates the rate of glacier ablation, with the rate of meltwater production from ice blocks covered with particles exceeding that from uncovered ice blocks. In our experiments, the rate of increase in meltwater production from particle-covered ice blocks was in the range 10–40 %. Under conditions of equal radiation intensity and particle mass, larger particle diameters resulted in a smaller rate of glacier meltwater production, with the difference in the

increase in volume of meltwater between different particle diameters in the range 9–25 mL. Under conditions of equal particle diameter and mass, higher radiation intensity led to a greater rate of glacier meltwater production, with the difference in the increase in volume of meltwater between different radiation intensities in the range 7–22 mL.

(2) Energy transfer between MDPs and glaciers constitutes the intrinsic foundation for accelerated ablation of glaciers, with the increase in the rate of glacier meltwater production serving as the outward manifestation of this energy transfer. A

functional relationship exists between the increase in particle diameter ($x$) and the corresponding increase in meltwater rate ($y$), which can be described by the function: $y = \alpha \cdot x^2$. With other conditions held constant, the base coefficient ($\alpha$) increases as irradiance increases and decreases as particle surface albedo increases. The relationship between irradiance ($w$) and the increase in meltwater rate ($y$) follows the function $y = \beta \cdot w + \gamma$. With other conditions held constant, the base coefficients ($\beta$ and $\gamma$) increase as particle diameter (<200 μm) increases and decrease as particle surface albedo increases. Particle surface albedo ($v$)

and the increase in meltwater rate ($y$) adhere to the function $y = \eta v + \delta$. With other conditions held constant, the base coefficients ($\eta$ and $\delta$) increase as irradiance decreases and increase as particle diameter increases. Additionally, the number of particles ($n$) and the increase in the glacier meltwater rate ($y$) have a functional relationship that can be expressed as $y = k \cdot n$, where the coefficient ($k$) is determined by the combination of particle diameter, irradiance, and surface diameter.

(3) For a glacier influenced by MDPs, given comprehensive understanding of the three critical parameters (i.e., particle

diameter, particle surface albedo, and irradiance), the theoretical increase in the rate of glacier melting effected by MDPs can be calculated using the normalized energy equation (Eq. 15), upon compiling statistics on particle quantity and types. It is essential to note that this equation is applicable when there is no particle stacking on the glacier surface. Additionally, if the wind speed over the glacier surface in the study area is notably high, it should also be taken into consideration.





## Appendix A

**Table A1.** Parameters of the thermophysical properties of the mineral rocks.

| Materials | Density (Kg m$^{-3}$) | Specific Heat Capacity (J kg$^{-1}$ K$^{-1}$) | Thermal Conductivity (W m$^{-1}$ K$^{-1}$) | Thermal Diffusivity ($\alpha\times10^{-6}$, m$^2$ s$^{-1}$) |
|---|---|---|---|---|
| Ice | 920 | 2100 | 2.21 | 1.14 |
| Basalt | 2840–2890 | 883.4–887.6 | 1.61–1.73 | 6.38–6.83 |
| Diabase | 3010 | 787.1 | 2.32 | 9.45 |
| Granite | 2500–2720 | 787.1–975.5 | 2.17–3.08 | 10.29–14.31 |
| Granodiorite | 2620–2760 | 837.4–1256.0 | 1.64–2.33 | 5.03–9.06 |
| Schist | 2700–2730 | 766.2–870.9 | 2.58–204 | 11.34–14.07 |
| Quartzite | 2680 | 787.1 | 6.18 | 29.52 |
| Marl clay | 2430–2640 | 778.7–979.7 | 1.73–2.57 | 8.01–11.66 |
| Travertine | 2530–2720 | 921.1–1000.6 | 2.52–3.79 | 10.75–14.97 |
| Limestone | 2410–2670 | 824.8–950.4 | 1.7–2.68 | 8.24–12.15 |
| Calcareous marl | 2430–2620 | 837.4–950.4 | 1.84–2.40 | 9.04–9.64 |
| Dense limestone | 2580–2650 | 824.8–921.1 | 2.34–3.51 | 10.78–15.21 |
| Mudstone | 2590–2670 | 908.5–925.3 | 2.32–3.23 | 9.89–13.82 |
| Argillite | 2620–2830 | 858.3 | 1.44–3.58 | 6.42–15.15 |
| Sandstone | 2350–2970 | 762–1071.8 | 2.18–5.1 | 10.9–423.62 |

*Note:* Data were obtained from the thermophysical properties of rocks as measured by the Chinese Academy of Sciences (2007 data) and reported in the ASHRAE Handbook (2019)

**Table A2.** Monitoring data of the National Aeronautics and Space Administration's Clouds and the Earth's Radiant Energy System project.

| Time period | Date | | | | | | | | | |
|---|---|---|---|---|---|---|---|---|---|---|
| | 2017 | | 2018 | | 2019 | | 2020 | | 2021 | |
| | 06.21 | 12.22 | 06.21 | 12.22 | 06.21 | 12.22 | 06.21 | 12.21 | 06.21 | 12.21 |
| 00-01 | 0 | 0 | 0 | 0 | 0 | 0 | 0 | 0 | 0 | 0 |
| 01-02 | 0 | 0 | 0 | 0 | 0 | 0 | 0 | 0 | 0 | 0 |
| 02-03 | 0 | 0 | 0 | 0 | 0 | 0 | 0 | 0 | 0 | 0 |
| 03-04 | 0 | 0 | 0 | 0 | 0 | 0 | 0 | 0 | 0 | 0 |
| 04-05 | 0 | 0 | 0 | 0 | 0 | 0 | 0 | 0 | 0 | 0 |
| 05-06 | 0 | 0 | 0 | 0 | 0 | 0 | 0 | 0 | 0 | 0 |
| 06-07 | 0 | 0 | 0 | 0 | 0 | 0 | 0 | 0 | 0 | 0 |
| 07-08 | 180 | 0 | 181 | 0 | 181 | 0 | 180 | 0 | 180 | 0 |
| 08-09 | 400 | 0 | 400 | 0 | 400 | 0 | 400 | 0 | 400 | 0 |
| 09-10 | 625 | 0 | 625 | 0 | 625 | 0 | 624 | 0 | 625 | 0 |
| 10-11 | 831 | 130 | 831 | 130 | 832 | 131 | 831 | 131 | 831 | 132 |
| 11-12 | 1006 | 313 | 1006 | 313 | 1006 | 313 | 1006 | 314 | 1006 | 314 |
| 12-13 | 1137 | 449 | 1137 | 450 | 1137 | 450 | 1137 | 450 | 1137 | 450 |
| 13-14 | 1215 | 529 | 1215 | 529 | 1215 | 529 | 1215 | 529 | 1215 | 530 |
| 14-15 | 1234 | 547 | 1234 | 547 | 1234 | 547 | 1234 | 547 | 1234 | 547 |



| Time period | Date | | | | | | | | | |
|---|---|---|---|---|---|---|---|---|---|---|
| | 2017 | | 2018 | | 2019 | | 2020 | | 2021 | |
| | 06.21 | 12.22 | 06.21 | 12.22 | 06.21 | 12.22 | 06.21 | 12.21 | 06.21 | 12.21 |
| 15-16 | 1195 | 501 | 1194 | 501 | 1194 | 500 | 1195 | 500 | 1195 | 500 |
| 16-17 | 1098 | 395 | 1098 | 394 | 1098 | 394 | 1098 | 394 | 1098 | 394 |
| 17-18 | 951 | 235 | 951 | 235 | 951 | 235 | 951 | 234 | 951 | 234 |
| 18-19 | 764 | 47 | 764 | 47 | 764 | 46 | 764 | 46 | 764 | 46 |
| 19-20 | 549 | 0 | 549 | 0 | 549 | 0 | 550 | 0 | 550 | 0 |
| 20-21 | 322 | 0 | 322 | 0 | 322 | 0 | 322 | 0 | 322 | 0 |
| 21-22 | 98 | 0 | 97 | 0 | 97 | 0 | 98 | 0 | 98 | 0 |
| 22-23 | 0 | 0 | 0 | 0 | 0 | 0 | 0 | 0 | 0 | 0 |
| 23-24 | 0 | 0 | 0 | 0 | 0 | 0 | 0 | 0 | 0 | 0 |

*Note:* The average irradiance within a 2-kilometer radius of Glacier #1 was calculated. The time periods in the table are presented in local time.

## Code and data availability

All data used in this study were generated through laboratory experiments conducted by the authors. The key parameters and results are presented in the main text. Since no external datasets or custom code were used, there is no external repository associated with this work.

## Author contributions

ZZ and XX designed the study. XX devised the methodology. XX implemented and modified experiment, collected MDPs data and produced the results. HS, GY and WG provided expert advice on the experiment. ZZ, XX and WW contributed to the analysis. XX wrote the manuscript with edits from the co-authors. ZZ, WW and XT provided funding for the field work.

## Competing interests

The contact author has declared that none of the authors has any competing interests.

## Acknowledgements

We would like to thank the engineers of the Iron Ore Project Department for their advice and guidance, and the workers who participated in the collection of specimens on sit.



**Financial support**

This research has been supported by the National Natural Science Foundation of China (grant no. 52204157), the National Natural Science Foundation of China (grant no. 2023D01C26) and the Third Xinjiang Scientific Expedition Program (grant no. 2022xjkk1000).

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
