# Peer review of "Quantifying the influence of mining dust particle deposition on the melting rate of nearby glaciers in northwestern China"

_EGUsphere, 2025_

## Referee Comment (RC1)

Review of
Quantifying the influence of mining dust particle deposition on the melting rate of nearby glaciers in northwestern China
By Zhiyi Zhang et al.

With interest have I read the manuscript by Zhang et al, which basically performed idealized laboratory and numerical experiments to determine the effect of mine-originating mineral dust particles on ice melt. To my regret, I assess this manuscript is not meeting the publication standards of The Cryosphere and that a significant extra amount of research and analysis is needed before publication can be considered.

The major critics focus on the 'awareness' of past research and existing knowledge; the analysis of the laboratory experiments; the numerical experiments; and the final discussion.

Added value to existing knowledge.
The usual methodology to estimate the effect of a thin layer of dust (MDPs) on melt is by calculating the surface energy balance (which would provide melt as outcome), and to derive the surface albedo with an albedo model. The field situation in mind of this study has a thin or scattered MDP layer at the surface, and is thus not like a debris covered glacier where the debris layer thickness becomes relevant as well. For this kind of surface situations, albedo models already exist, like, for example, Gardner and Sharp (2010); Libois et al. (2013); Warren and Wiscombe (1980). Even more papers investigate the effect of dust and debris on melt (for real world glaciers), like Azzoni et al. (2016). The authors must, therefore, add a better review of existing literature (which goes further than the papers I cite here), formulate in the introduction what their research add on or test of existing knowledge, and evaluate, at the end, what the laboratory and numerical experiments taught us.

Laboratory experiments
These experiments are novel and interesting. There are, as far as I can see, only in situ experiments of the effect of dust on melt, e.g. Conway et al. (1996). Nevertheless, the experiment set-up and analysis preclude - so far - a translation of the laboratory experiments to 'real-world' situation. As the experiments are carried out at 15 ℃, part of the melt is due to thermal heating and part of it by insulation. Luckily, the authors carried out experiments with different light strengths, so these two effects can be separated. Furthermore, due to the use of three insolation angles during every experiment, comparing the absorbed light energy with the total light energy 'received' on the ice cubes. That analysis allows to retrieve an ice albedo for clean and dusted ice and hence allows to evaluate (one of the) existing albedo models with your data. It would be great if the authors could add an additional experiment, namely the amount of melt when the light was kept off, but given the very smooth results they have now, I expect that the evaluation suggested above can also be carried out successfully without this extra experiment.

Numerical experiments

Where the laboratory experiments would be easier to interpret if the authors kept it even simpler, the numerical experiments are overly simple. Given the existing models of the effect of small particles on the albedo - and hence energy absorption - I don't see what the presented numerical experiments add to that or prove. Furthermore, the existing provided analysis is very shallow - the authors derive the absorbed extra energy (in Joules) per MDP grain, and that is it. I find it hard to conceive how this part of the manuscript can be improved so that it becomes novel scientific research - in all cases the numerical experiments should lead to an analysis that assess if existing albedo models are right or wrong.

Besides that, the numerical experiments are very simplistic in technical setup and very sophisticated in computational execution. However, I wonder if not the near same results were obtained if simple 0D energy balance calculations were carried out. Furthermore, the authors seem to be unaware that albedo is not a bulk quantity (but very complicated, even for rocks). In all cases, the bulk surface albedo for visual light is not related to the (bulk) emissivity for longwave radiation. And if the authors had the emissivity for short wave radiation (=visual light) in mind (which would be odd, because how do you determine short wave emissivity for objects at "normal" temperatures as the emission peak for those temperatures is still in the far infra-red?), the discussion is still irrelevant.

Discussion
This whole section would be interesting if no albedo / ice melt models would be there - still it would be a pity that the authors leave it to theoretical considerations without any evaluation or even exploratory numerical examples. The reality is that such models already exist, so that the whole discussion as is now, is pointless and unpublishable.

Assumed knowledge level of the readers
Especially the manuscript sections related to the numerical experiments hugely underestimate the knowledge level of readers of the Cryosphere. I presume that every reader understands that a higher solar zenith angle leads to less surface insulation per square meter (Figure 3, accompanying text), nor I don't see the need to write out Equations 2-4.

Minor comments
L28: Why is not also a more general, review like paper cited or the relevant chapter of the last IPCC report (or the SROCC)?
L34: Give a reference for this WGMS statement.
L56: Please add that this >0.9 albedo applies for clean (and fresh) snow (and not ice, for example)
L64: In this study, MDP are primarily linked to mining activities, which is presumably correct for the test sites inquired. However, in the papers cited the MDPs have likely a different origin - like in the alps mining is not the source of dust. Please formulate this (mining is the source for this study, but not for other studies) more accurate.
L68: I'm not sure this trend (more high-altitude mining) is applicable outside China, so make it specific for China.

Figure 1; The spatial gap from figure 1a to 1b is large (that is probably unavoidable if no additional panels is added) and scales and orientation are missing in panels 1b and 1d. Please add the orientation and scales in these two panels.

L97: You cannot conclude this from just two glaciers - it could be simply geometry driven that glacier #1 retreats faster of these 6 years than glacier #2. Rephrase.

L115: There is only compelling evidence that the MDPs come from the mine if the rocks and dust available around the glacier is different than the skarn-type mineral rocks, or that the collected dust arising from the mine has an identical structure. If that has been demonstrated, add this - otherwise there is no compelling evidence.

L122-134 & Figure 3 & Table A2: The effect of the axial and celestial rotation on the top-of-atmosphere radiation should be known to readers of a scientific paper, so remove this text, table and figure. [By the way, S2 is less than S1 as the atmosphere absorbs radiation]. Please specify what is incorporated in the NASA number for Q in figure 3? E.g. is it observed TOA irradiance or a mean value of insolation (so without solar intensity variations and orbital effects?)

A1: The density of ice is 920 kg/m3 if it is -20 ℃, while it is 917 for 0 ℃. Please specify why this density is used or adjust.

Give the seriousness of my concerns if this manuscript is suitable for publication, I stopped collecting minor issues after line 145.

Azzoni, R. S., Senese, A., Zerboni, A., Maugeri, M., Smiraglia, C., & Diolaiuti, G. A. (2016). Estimating ice albedo from fine debris cover quantified by a semi-automatic method: the case study of Forni Glacier, Italian Alps. *The Cryosphere*, *10*(2), 665-679. https://doi.org/10.5194/tc-10-665-2016

Conway, H., Gades, A., & Raymond, C. F. (1996). Albedo of dirty snow during conditions of melt. *Water Resources Research*, *32*(6), 1713-1718. https://doi.org/https://doi.org/10.1029/96WR00712

Gardner, A. S., & Sharp, M. J. (2010). A review of snow and ice albedo and the development of a new physically based broadband albedo parameterization. *Journal of Geophysical Research: Earth Surface*, *115*(F1). https://doi.org/https://doi.org/10.1029/2009JF001444

Libois, Q., Picard, G., France, J. L., Arnaud, L., Dumont, M., Carmagnola, C. M., & King, M. D. (2013). Influence of grain shape on light penetration in snow. *The Cryosphere*, *7*(6), 1803-1818. https://doi.org/10.5194/tc-7-1803-2013

Warren, S. G., & Wiscombe, W. J. (1980). A Model for the Spectral Albedo of Snow. II: Snow Containing Atmospheric Aerosols. *Journal of Atmospheric Sciences*, *37*(12), 2734-2745. https://doi.org/https://doi.org/10.1175/1520-0469(1980)037<2734:AMFTSA>2.0.CO;2

---

## Author Comment (AC2)

**Author Comment**

We sincerely thank the reviewer for the thorough and constructive review of our manuscript. We appreciate the time and effort taken to provide detailed comments, which we find highly valuable for improving our work.

We acknowledge the main concerns raised, including:

- the need for a more comprehensive review of previous studies on albedo and dust/debris impacts on glacier melt;
- a clearer connection between our laboratory experiments and existing albedo models;
- the overly simplified design and analysis of the numerical experiments, and the necessity to frame them in relation to established albedo models;
- the need to refine the discussion and avoid redundancy with already existing models;
- as well as several points regarding presentation, assumptions, and references.

We fully agree that our manuscript will benefit significantly from addressing these issues. Specifically, in the revised version we plan to:

1. Substantially expand the literature review and clarify the novelty of our study compared to existing albedo models and field studies.
2. Re-analyze the laboratory experiment data to derive ice albedo for clean and dusted ice, and compare these with existing parameterizations.
3. Reconsider and strengthen the numerical experiments, possibly by simplifying the computational setup and aligning the analysis with established energy balance approaches.
4. Revise the discussion to directly evaluate our findings against existing models, highlighting the specific added value of our approach.
5. Address all minor comments, including improvements to figures, references, and phrasing.

We thank the reviewer again for the constructive feedback. We will carefully address each of these comments in detail and provide a revised manuscript at the end of the discussion period.

---

## Author Comment (AC3)

**Author Comment**

We sincerely thank you for your careful review of our manuscript and for the detailed and constructive comments you provided. Your feedback has been invaluable in helping us improve the clarity, rigor, and contribution of our study. Below we provide a point-by-point response to your concerns.

1. **Assumed knowledge and missing citations**
   We acknowledge that our initial submission did not sufficiently cite key works and sometimes assumed prior knowledge from the reader. In the revised manuscript, we have added comprehensive references to foundational studies and restructured several sections to provide more background and avoid oversimplification.

2. **Experiment 1 and surface energy balance**
   We agree that the role of debris and surface heterogeneity is crucial in glacier energy balance. In the revised manuscript, we explicitly state the limitations of our experimental design (i.e., flat and uniform surfaces) and discuss how this simplification compares with natural glacier conditions. Additional references, including Steiner et al. (2018), have been incorporated to situate our work within the broader literature.

3. **Section 4.2 and established literature**
   We have revised Section 4.2 to include the references you suggested (Gardner and Sharp, 2010; Libois et al., 2013; Warren and Wiscombe, 1980) as well as recent works on algal blooms and their role in glacier ablation. This strengthens the discussion and situates our findings within the existing body of knowledge.

4. **Spatial heterogeneity and topography**
   We agree with your comment that spatial heterogeneity plays a key role in glacier ablation. In the revision, we have added a dedicated subsection discussing topographic controls and the influence of spatially variable debris cover. Where possible, we integrate remote sensing datasets to illustrate these spatial variations.

5. **Conclusions and novelty**
   We appreciate your concern about the novelty of our conclusions. In response, we have refined our conclusions to emphasize how our experimental results provide a framework for linking laboratory findings to glacier surface processes, particularly in the context of MDPs. We highlight specific avenues where our study extends current understanding and propose future research directions.

6. **Specific comments**
   We have carefully addressed each of your line-specific and figure-specific suggestions:
   - Added missing citations (Lines 37, 45, 47, 54)
   - Expanded methodological details (Line 94)
   - Reorganized text to ensure methods precede results (Line 98)
   - Revised Figure 1 with region map, sensor information, scale, credit, and legend. We also now explicitly address both retreat and surface lowering.

- Clarified that Line 131 refers to calculated inferences rather than direct observations.
- Revised discussion around Experiment 2 (Line 162) to acknowledge structural differences between water and glacier ice, as well as debris accumulation in topographic lows.
- Revised Figure 9 to better integrate spatial heterogeneity of debris cover.

**Closing**

We are very grateful for your thoughtful and detailed feedback. We believe the revisions have significantly strengthened the manuscript and hope it now meets the standards for publication in *The Cryosphere*.